# Citrullination of glucokinase is linked to autoimmune diabetes

Mei-Ling Yang [1], Sheryl Horstman[2], Renelle Gee[1], Perrin Guyer[2], TuKiet T. Lam [3,4], Jean Kanyo[4], Ana L. Perdigoto [5], Cate Speake [6], Carla J. Greenbaum [6], Aïsha Callebaut[7], Lut Overbergh[7], Richard G. Kibbey[5,8], Kevan C. Herold [5,9], Eddie A. James [2] & Mark J. Mamula [1✉]

Inflammation, including reactive oxygen species and inflammatory cytokines in tissues amplify various post-translational modifications of self-proteins. A number of post-translational modifications have been identified as autoimmune biomarkers in the initiation and progression of Type 1 diabetes. Here we show the citrullination of pancreatic glucokinase as a result of inflammation, triggering autoimmunity and affecting glucokinase biological functions. Glucokinase is expressed in hepatocytes to regulate glycogen synthesis, and in pancreatic beta cells as a glucose sensor to initiate glycolysis and insulin signaling. We identify autoantibodies and autoreactive CD4+ T cells to glucokinase epitopes in the circulation of Type 1 diabetes patients and NOD mice. Finally, citrullination alters glucokinase biologic activity and suppresses glucose-stimulated insulin secretion. Our study define glucokinase as a Type 1 diabetes biomarker, providing new insights of how inflammation drives post-translational modifications to create both neoautoantigens and affect beta cell metabolism.

[1] Section of Rheumatology, Allergy and Immunology, Department of Internal Medicine, Yale University, New Haven, CT, USA. [2] Center for Translational Immunology, Benaroya Research Institute at Virginia Mason, Seattle, WA, USA. [3] Department of Molecular Biophysics & Biochemistry, Yale University, New Haven, CT, USA. [4] Keck MS & Proteomics Resource, WM Keck Foundation Biotechnology Resource Laboratory, New Haven, CT, USA. [5] Section of Endocrinology, Department of Internal Medicine, Yale University, New Haven, CT, USA. [6] Center for Interventional Immunology, Benaroya Research Institute at Virginia Mason, Seattle, WA, USA. [7] Laboratory for Clinical and Experimental Endocrinology, KU Leuven, Leuven, Belgium. [8] Department of Cellular & Molecular Physiology, Yale University, New Haven, CT, USA. [9] Department of Immunobiology, Yale University, New Haven, CT, USA. ✉email: mark.mamula@yale.edu

Type 1 diabetes (T1D) is characterized as an insulin-dependent glucose metabolic disorder arising from inflammation of the pancreatic islets and amplified by autoreactive autoimmune B and T lymphocyte responses. Local inflammatory cytokines and reactive oxygen amplify post-translational protein modifications (PTMs), including deamidation, oxidation, carbonylation and citrullination, all capable of compromising immune tolerance[1]. In particular, citrullination has been identified in a variety of tissues and studied extensively as biomarkers of rheumatoid arthritis (RA)[2]. Anti-cyclic citrullinated peptide antibodies (anti-CCP) and T cells arise early in RA, correlate with disease severity, and are now routinely used for the diagnosis of RA[3–6]. An emerging group of PTM biomarkers important in T1D have been identified including islet cell auto-antigen 69 (ICA69), insulin, glutamic acid decarboxylase 65 (GAD65), islet antigen-2 (IA-2) and zinc transporter 8 (ZnT8). Citrullinated-78-kDa glucose-regulated protein (GRP78) and -GAD65 were found to elicit vigorous B and T cell autoimmune responses in both human T1D and NOD murine disease[7–9].

Glucose homeostasis in humans is highly regulated by the activity of glucokinase, catalyzing glucose phosphorylation, a first rate-limiting step of glycolysis in the liver and pancreas[10]. More than 600 mutations of the human glucokinase gene have been described, resulting in reduced glucokinase enzyme activity in pancreatic β-cells and in hepatocytes[11]. As previously described, patients with maturity onset diabetes of the young (MODY) is linked to specific mutations of the glucokinase gene, classified as MODY2[12,13]. Clinical trials of glucokinase activators have been investigated in patients with type 2 diabetes (T2D), including piragliatin, MK-0941, AZD1656[14] and dorzagliatin[15]. In contrast to T2D and monogenic diabetes, much less is known regarding the role of glucokinase dysfunction in autoimmune type 1 diabetes (T1D).

Citrullinated proteins are the product of the conversion of arginine to citrulline, catalyzed by peptidylarginine deiminases (PADs) in the presence of $Ca^{++}$ [16]. There are five PAD enzymes (PAD 1–4 and 6) and levels are often increased with tissue inflammation[17]. Among these, PAD2 is widely expressed in different tissues such as brain, spinal cord, spleen, pancreas, bone marrow, skeletal muscle but not detected in liver, kidney and testis[18,19]. PAD2 expression and activity is increased in synovial fluid from anti-CCP positive RA patients and positively correlated with disease activity[20]. Relevant to T1D, transcriptomic and proteomic profiling demonstrates that PAD2 is highly expressed in prediabetic nonobese diabetes (NOD) islets[21]. In this work, we demonstrate the presence of autoimmune B and T cells to glucokinase epitopes in patients with T1D. Moreover, we report that inflammation causes glucokinase to undergo citrullination almost exclusively in the pancreas, but not in the liver of NOD mice. Citrullination alters the enzyme kinetics of pancreatic glucokinase and PAD2/PAD4 inhibitor partially restores impaired insulin secretion mediated by proinflammatory cytokines. The work suggests that citrullination of glucokinase is a marker of beta cell dysfunction as well as an autoimmune biomarker of T1D and support a potential therapeutic strategy of inhibiting PAD enzymes to restore beta cell metabolic pathways.

## Results

### GK citrullination in inflamed NOD islets and inflammatory stressed-beta cells.
We first performed immunohistochemistry staining and confocal microscope to examine if protein citrullination is increased in islets from the spontaneous NOD murine model of T1D. NOD mice develop hyperglycemia, insulitis, and lymphocyte infiltration of the pancreatic islets as one model of human T1D. Protein citrullination in islets is already observed in 3.5 week old NOD mice without infiltration of lymphocytes (Fig. 1a, b) and increases in citrullination staining intensity with increasing age of the mice (Fig. 1a). Increased citrulline staining coincided with insulitis, as confirmed by hematoxylin–eosin (H&E) staining and anti-CD3 staining from 16 week old NOD mice (Fig. 1b). Overall protein citrullination is not detected in liver and pancreas extracts from age matched C57Bl/6 mice or in NOD liver, but is significantly increased in NOD pancreas (Fig. 1c). As one band of the extract matched the molecular weight of GK, PAD treated human GK served as a citrulline staining control.

Glucokinase is the primary glucose sensor since even small fluctuations of its enzyme activity alter the threshold of glucose-stimulated insulin secretion in pancreatic β-cells[10,22]. In liver, the major role of glucokinase is to regulate the glycogen synthesis. As previously reported[23] and confirmed here, GK levels are expressed in significantly higher levels relative to total protein in liver compared to pancreas from both C57Bl/6 and NOD mice (Fig. 1d). We next immunoprecipitated GK with specific antibody to confirm GK citrullination in NOD pancreas. As expected, we captured significantly more glucokinase from NOD liver extracts compared to pancreas from 16 week old NOD mice (Fig. 1e). In contrast, the citrullinated glucokinase signal is significantly increased in pancreas compared to liver from NOD mice (Fig. 1e). These data suggest that little overall GK citrullination occurs in the liver, while significant pancreatic GK citrullination arises in the same NOD mouse as disease progresses.

To assess whether inflammation triggers GK citrullination in insulin secreting β-cells, a cocktail of proinflammatory cytokines (rmIFNγ and rhIL-1β) were used to model the insulitis using rat insulinoma cell line, INS-1 cells, which displays the key characteristics of pancreatic β cells. The level of protein citrullination was increased in cytokine-treated INS-1 cells over time (12, 24 or 48 h) by immunofluorescence and by flow cytometry (Supplementary Fig. 1a, b, respectively). As expected, $GK^+$/citrullination$^+$ populations of INS-1 cells were also increased after cytokine treatment compared to untreated cells in a time dependent manner (Supplementary Fig. 1c). Finally, cytokine triggered glucokinase citrullination in INS-1 cells was confirmed by immunoprecipitation with anti-glucokinase and immunoblot with anti-peptidyl-citrulline (Supplementary Fig. 1d). In addition, we incubated islets from non-diabetic human organ donors with proinflammatory cytokines (IFNγ alone or IFNγ and TNFα) and assessed the induction of citrullinated GK. Our results showed that citrullinated modified-GK was significantly promoted in human islets treated with IFNγ alone (Supplementary Fig. 1e) compared to untreated human islets. Collectively, these results illustrate overall pancreatic citrullination and that citrullination of GK can be specifically amplified by inflammatory cytokines in tissue, islets, and cell based systems.

### Autoantibodies against GK and citrullinated GK arise in both murine and human T1D.
We first determined if citrullinated GK may be a neoantigen in triggering autoreactivity in T1D. NOD mouse serum recognized both rhGK and PAD-treated rhGK proteins by immunoblot, while serum from control B10.BR mice did not (Fig. 2a). We next examined IgG autoantibodies to both GK and citrullinated GK by solid phase immunoassay (ELISA) in the serum of NOD mice. Anti-GK and anti-citrullinated-GK IgG levels were significantly higher in both pre-diabetic NOD mice ($n = 78$) and diabetic NOD mice ($n = 16$; blood glucose content greater than 250 mg/dl) compared to control, B10.BR mice ($n = 52$) (Fig. 2b). Subdividing the samples by age groups clearly show that anti-GK and anti-citrullinated-GK IgG arise early in

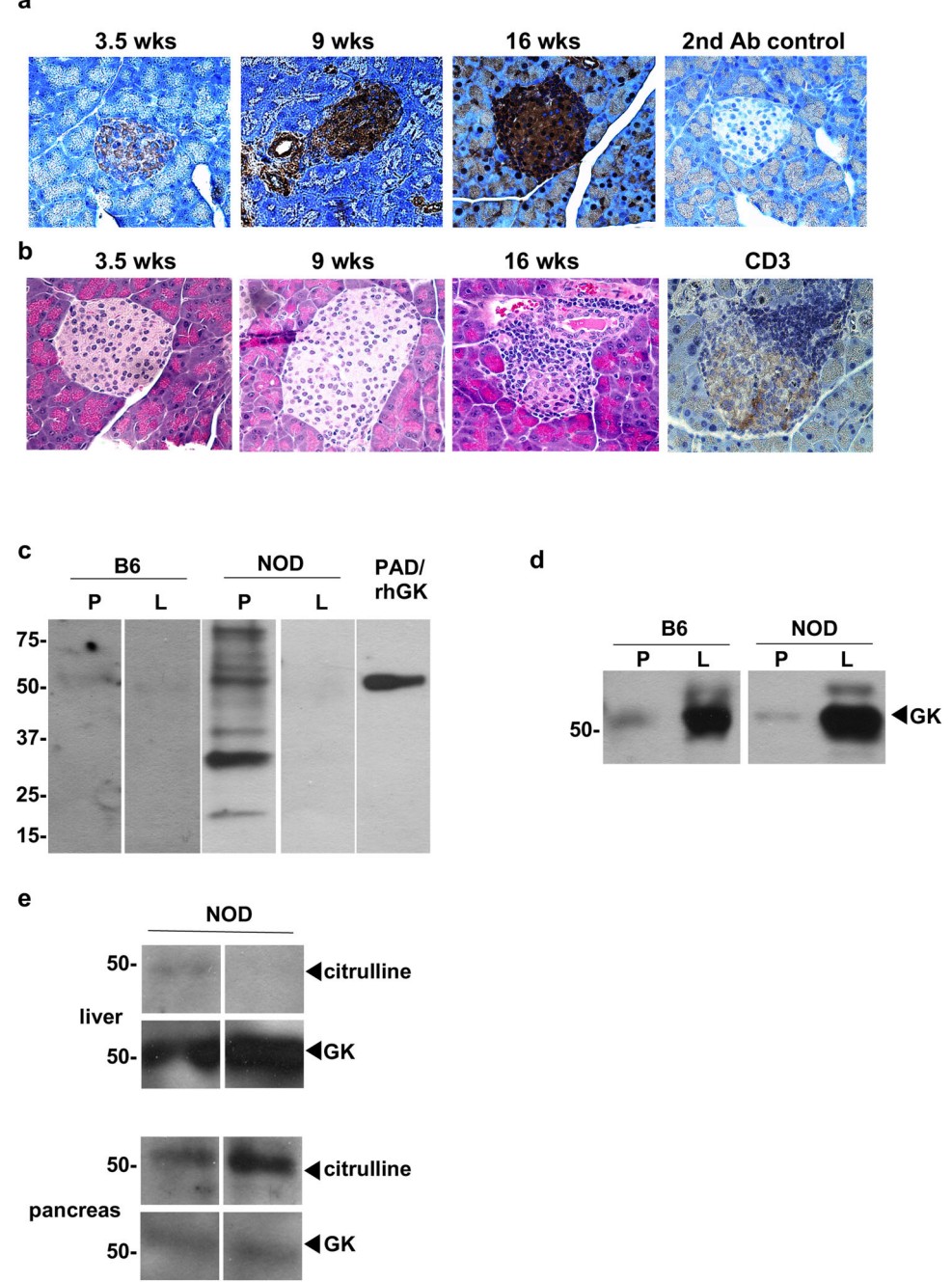

**Fig. 1 Citrullination of glucokinase in NOD pancreas, but not in liver. a** Representative immunodetection of citrullinated proteins in NOD pancreas tissue sections at different ages of NOD mice (between 3.5 and 16 weeks of age). Brown staining demonstrates positive immunohistochemical reaction for citrulline with anti-citrulline antibody. **b** Representative H&E of NOD pancreas tissue sections at different ages (between 3.5 and 16 weeks of age) and anti-CD3 at 16 weeks of age. **c**, **d** Representative tissue extracted proteins, 50 μg per lane, from liver (L) or pancreas (P) of C57Bl/6 (B6) and NOD mice (16 weeks old) were separated on SDS-polyacrylamide gel electrophoresis (PAGE), and protein spots were stained by anti-modified citrulline detection kit (**c**) and anti-GK (**d**). PAD-treated recombinant human GK (PAD-rhGK) serves as the positive control for protein citrullination (**c**). **e** The tissue extracted proteins from NOD mice (16 weeks old) were immunoprecipitated using anti-GK and immunoblotted with anti-peptidyl citrulline and anti-GK as indicated.

NOD mice, by 4–6 weeks of age, and before the onset of hyperglycemia. The autoantibody levels against GK and citrullinated GK continue to increase up to 20 weeks of age in NOD mice (Supplementary Fig. 2a, b). However, congenic non-obese-resistant (NOR) mice, lacking insulitis and diabetes, do not produce IgG autoantibodies to either GK or citrullinated GK (Supplementary Fig. 2c).

Serum autoantibodies against citrullinated GK and its native counterpart in individuals with T1D ($n = 55$) and healthy control subjects ($n = 18$) were examined by ELISA. As shown in Fig. 2c, patients with T1D had significantly higher anti-GK and anti-citrullinated-GK IgG levels compared to healthy subjects. Other established autoantibodies were also screened within the same set of human serum - including anti-insulin, anti-GAD65, anti-IA2 and anti-ZnT8. Interestingly, anti-GK and anti-citrullinated-GK antibodies were statistically correlated with the presence of anti-IA2 autoantibodies, but not statistically linked with anti-insulin, anti-GAD65 or anti-ZnT8 (Supplementary Table 1).

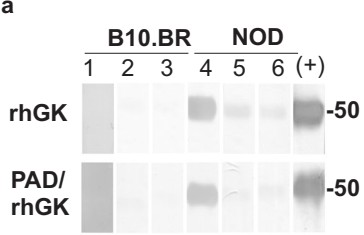

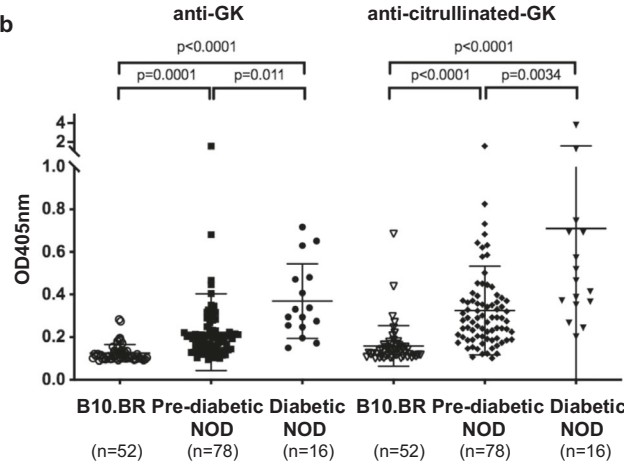

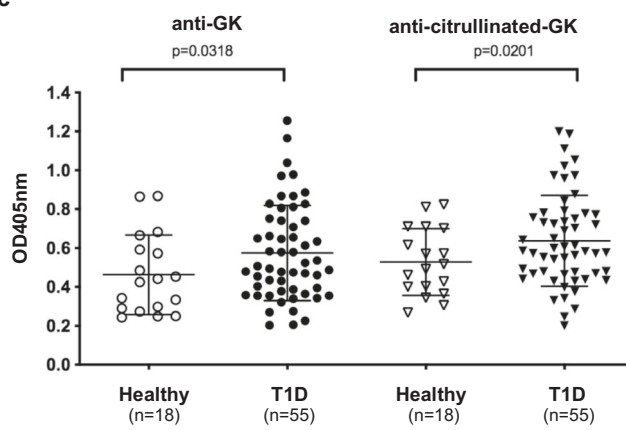

**Fig. 2 The prevalence of autoantibodies against glucokinase and citrullinated glucokinase in both murine and human models of T1D.**
**a** rhGK and PAD-rhGK (1 μg per lane) were subjected to electrophoresis and followed by immunoblot with rabbit anti-GK as positive control (+) and sera from 16-week-old B10.BR and NOD mice, respectively (lanes 1–6). **b–c** The serum levels of anti-GK and anti-citrullinated GK from NOD mice and control mice (**b**) and from patients with T1D and healthy subjects (**c**) were measured by ELISA (as described in Methods). The number of samples analyzed is indicated at the bottom of each group and the error bars indicated SD of mean. Statistical analysis was performed using Student's *t*-test.

**CD4+ T cells specific for citrullinated GK epitopes in the circulation of T1D patients.** We next investigated whether glucokinase peptides are recognized by autoreactive CD4 T cells in patients with T1D. A discrete number of HLA class II haplotypes are highly associated with T1D *susceptibility*[24]. In particular, *HLA-DRB1\*04:01 (DR0401), DQA1\*03:01-DQB1\*0302 (DQ8), HLA-DRB1\*03:01 (DR0301) and DQA1\*05:01-DQB1\*02:01 (DQ2)*, confer high risk with about 80-90% of T1D patients carrying at least one of these risk alleles[25]. Prior studies documented that citrullination can enhance peptide binding to

**Table 1 Sequences and binding affinities for antigenic glucokinase peptides.**

| Peptide | Amino acid sequence[a] | IC50[b] (μM) |
|---------|------------------------|--------------|
| GK 192  | RGD**FEMDVVAMV**NDT     | 0.35         |
| GK 199  | VVA**MVNDTVATM**ISCY    | 2.5          |
| GK 266  | LDE**FLLEYDRLV**DES     | 5.6          |
| GK 270  | LLE**YDRLVDESS**ANP     | 2.0          |
| GK 346  | KQIYNI**LSTLGL[Cit]PS** | 0.47         |

[a]The predicted minimal epitope is bolded in each sequence. Possible secondary motifs are underlined.
[b]IC50 represents the peptide concentration that displaces half of the reference peptide.

DR0401, generating GAD65 neo-epitopes[26]. Therefore, we sought to identify glucokinase sequences that can be bound and presented by DR0401 by scoring all possible nine amino acid motifs within GK using a previously published algorithm[6,27]. For these predictions, Arg residues that indicated by mass spectrometry data to be accessible for modification by PAD enzyme were replaced by citrulline (Supplementary Fig. 3). A total of 38 peptides were synthesized and their binding to recombinant DR0401 protein was assessed using a competition bioassay[6,26]. Ten candidate peptides that bound to DR0401 with appreciable affinity (Supplementary Table 2). DR0401 tetramers for those peptides were produced and used these to investigate the ability of each peptide to elicit CD4+ T cell responses in vitro. PBMCs from subjects with T1D were stimulated with pools of glucokinase peptides for two weeks, and subsequently stained with the corresponding individual tetramers, revealing five peptides that elicited detectable populations of tetramer-positive T cells in multiple subjects (Supplementary Fig. 4). The sequences of these immunogenic peptides and their predicted motifs are summarized in Table 1. Tetramer-positive T cell clones were isolated for all indicated peptides, further confirming T cell recognition of these GK epitopes (Supplementary Fig. 5).

Among the five immunogenic peptides, three had an Arg residue within the predicted minimal motif (GK 266, GK 270, and GK346) but only one of these, GK residue 358, was shown to be citrullinated. To investigate whether citrullination influenced the binding of these GK epitopes, HLA binding of citrulline was compared to Arg containing versions of the GK peptides (Supplementary Table 3). The results indicated that citrullination did not significantly alter binding for any of the peptides, suggesting that the citrulline/Arg residues are not involved with HLA binding. Consistent with this observation, the predicted minimal motifs (underlined in Table 1) place the citrulline/Arg residues for GK266, GK270, and GK346 in positions (p5, p-1, and p10, respectively) that are known to be potential T cell receptor contact residues.

To enumerate GK-reactive T cells in human subjects, we next applied a direct tetramer enrichment approach to measure their frequency in the peripheral blood of subjects with T1D and HLA matched controls[28]. Individual GK peptides were used to prepare HLA class II tetramers labeled with either PE, PE-CF594, or PE-Cy5, staining two separate aliquots of cells to assess the five GK epitopes simultaneously (representative results shown in Supplementary Fig. 6). Some GK-reactive CD4+ T cells were present in controls. However, the combined frequency of GK-reactive CD4+ T cells in subjects with T1D was significantly higher than in controls (*p* = 0.0007) (Fig. 3a). Utilizing CD45RA and CCR7 as markers to distinguish antigen-experienced (CD45RA-) versus naïve T cells (CD45RA+CCR7+), healthy subjects tended to have a higher proportion of GK-reactive T cells that were naïve than subjects with T1D (*p* = 0.06) (Fig. 3b). Thus, subjects with established T1D had significantly greater numbers of GK-reactive

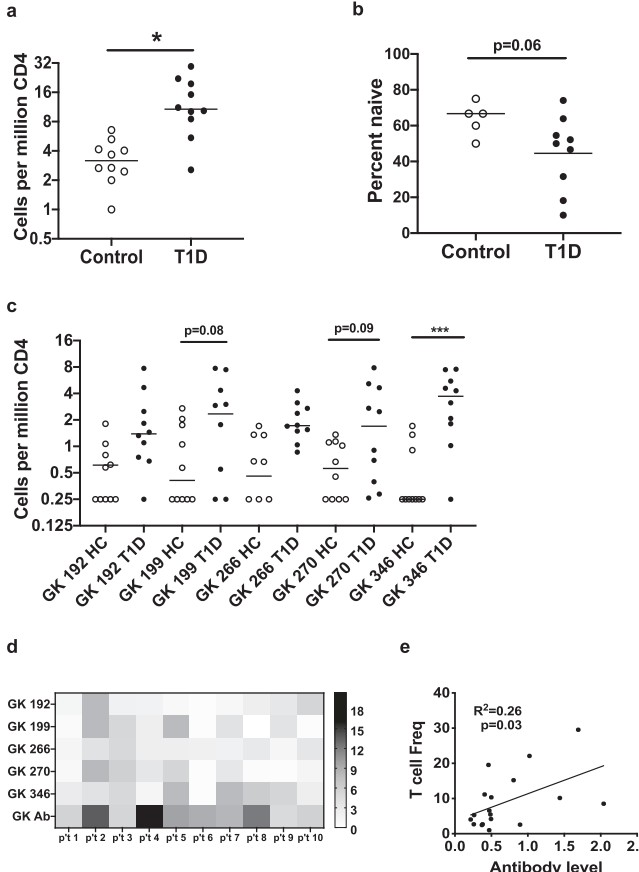

**Fig. 3 GK specific CD4⁺ T cells are more frequent in subjects with T1D and are correlated with anti-GK antibodies. a** Comparison of total GK reactive CD4⁺ T cell frequencies in the peripheral blood of 10 healthy controls and 10 subjects with T1D. The total frequencies of glucokinase-reactive T cells (GK 192, 199, 266, 270, and 346 combined) were significantly higher (*; $p = 0.0007$, Mann Whitney test) in subjects with T1D (black circles) than in HLA matched controls (white circles). **b** GK specific CD4⁺ T cells tended to be more naïve (CD45RA+ CR7+) in healthy controls than in subjects with T1D ($p = 0.064$, Mann Whitney test). **c** Individual frequencies were significantly higher in subjects with T1D (black circles) than in healthy controls (white circles) for GK 346 (***; $p = 0.0004$, ANOVA followed by Sidak's multiple comparisons test). Individual frequencies trended toward being higher for GK 199 and GK 270 ($p = 0.08$ and $p = 0.09$ respectively) but did not reach statistical significance. **d** A heatmap analysis of individual GK T cell frequencies in subjects with T1D, indicating patterns of reactivity that differed between subjects. In the heatmap each column represents one subject and each row reflects one GK specificity, with each square color coded to indicate the observed T cell frequency in cells per million. The antibody readout is indicated at the bottom individually. **e** There was a significant positive correlation between the combined frequency of GK-reactive T cells and levels of anti-glucokinase antibodies ($p = 0.030$, simple linear regression).

CD4⁺ T cells than controls and their GK-reactive T cells tended to show increased markers of antigen exposure. GK 346 specific T cells were significantly more frequent in subjects with T1D than in controls ($p = 0.0004$), but GK 199 and GK 270 also trended toward having higher frequencies in subjects with T1D ($p = 0.08$ and $p = 0.09$, respectively) (Fig. 3c). T cell specificity data was expressed as a heat map to visualize patterns of glucokinase-specific T cells in different individuals (Fig. 3d). From this analysis it was evident that some subjects had high frequencies for multiple epitopes (e.g., T1D #2 and T1D #5) whereas others had

very few glucokinase-specific T cells (e.g., T1D #6). However, T1D #2, #5 and #6 all have detectable anti-glucokinase IgG (indicated at the bottom of Fig. 3d). We next asked if the frequency of glucokinase-reactive T cells may be correlated with characteristics such as disease duration, age at diagnosis biological sex, or levels of glucokinase specific antibodies. Considering both T1D patients and controls, we observed a significant positive correlation between the combined frequency of glucokinase-reactive T cells and levels of anti-glucokinase antibodies (Fig. 3e). No other phenotypic associations reached significance.

**The effect of citrullination on the biological function of GK in beta cells.** Proinflammatory cytokines are known to attenuate glucose stimulated insulin secretion (GSIS), mediated by glucokinase, as studied elsewhere[29–31]. We utilized a subclonal cell line from the parenteral INS-1 cells, INS-1E, to examine GSIS and how citrullination may alter the glucose sensing/insulin secreting metabolic pathway. We also utilized YW3-56, a pan-PAD inhibitor (IC50: 5.5–1 μM and 1–5 μM for PAD2 and PAD4, respectively), in studies of GSIS in INS-1E cells as a means of blocking protein citrullination. First, we confirmed by immunoblot detection of citrulline that IFNγ+ IL-1β cytokine-induced citrullination levels were reduced in the presence of YW3-56 in INS-1E cells (Fig. 4a). As shown in the left panels of Fig. 4b, INS-1E cells secrete insulin in response to glucose in a dose-dependent manner (2.5, 5 and 16.7 mM of D-(+)-glucose, white bar), while IFNγ + IL-1β cytokines diminished insulin secretion upon glucose stimulation (black bars). As expected, YW3-56 alone did not affect citrullination levels or GSIS in INS-1E cells (Fig. 4a, b, striped bar). Of note, YW3-56 can partially correct IFNγ + IL-1β suppressed GSIS at 16.7 mM glucose when YW3-56 was cocultured with cytokines for 48 h (grey bar in left panel of Fig. 4b). The use of pyruvate stimulation of beta cells will bypass gluco-kinase to stimulate insulin secretion[30]. Pyruvate stimulated insulin secretion was also found to decrease under IFNγ+ IL-1β exposure (the right panel of Fig. 4b). Moreover, YW3-56 was also able to correct cytokine-mediated suppression of insulin secretion upon pyruvate stimulation. However, insulin secretion to depolarization with KCL was not affected by cytokines in the presence or absence of YW3-56 (the middle panel of Fig. 4b). In toto, our data suggest that IFNγ + IL-1β−triggered citrullination affects not only glycolysis (mediated by glucokinase) but also down-stream metabolism from glycolysis in the insulin biosynthesis pathway. However, there appears to be no effect of citrullination on the depolarization step of insulin secretion in beta cells.

Recombinant human pancreatic glucokinase (rhGK) was used to map potential citrullination residues and determine if citrullination alters glucokinase catalytic activity. Full length rhGK was incubated with PAD in the presence of $Ca^{++}$. Citrullinated rhGK migrates slightly slower on SDS-PAGE due to the loss of positive charge as shown in the left panel of Fig. 5a. Citrullination of GK was also confirmed by immunoblot (the right panel of Fig. 5a). Mass spectrometry identified 11 citrullination modification residues in PAD-treated rhGK (Supplementary Fig. 3 and all citrullinated peptides identified in PAD-treated rhGK are listed in supplementary Table 6). Thus, 34% of Arg residues (11 Arg out of total 32 Arg residues) were available to be citrullinated in vitro by PAD. As shown in Fig. 5b, citrullination of rhGK by PAD in vitro was found to reduce the catalytic activity of the enzyme by 20%. Next, we subjected the native rhGK and PAD-treated rhGK to steady-state kinetic analyses at 37 °C with various concentration of substrate (glucose). PAD-treated rhGK demonstrated relatively similar Vmax compared to native rhGK (119.9 ± 7.7 μmol/mg/min of PAD-treated rhGK versus 115.1 ± 10.5 μmol/mg/min of native

**a**

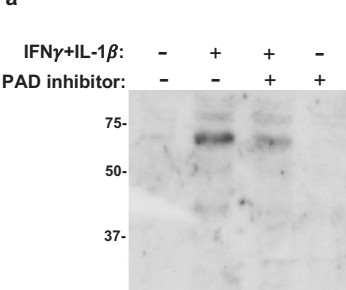

**b**

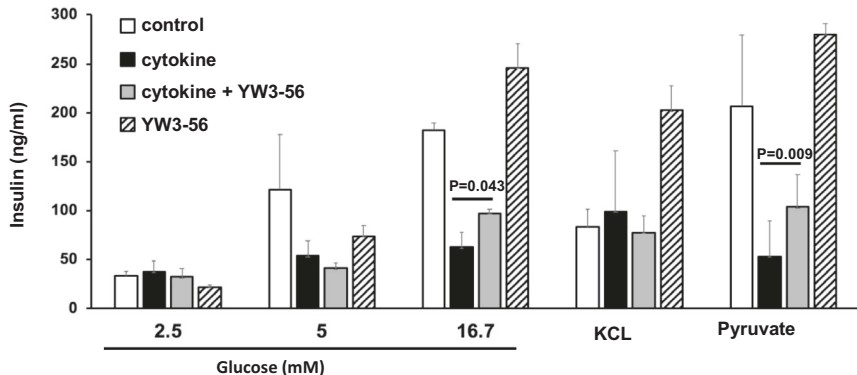

**Fig. 4 PAD inhibitor reduces overall citrulline modification and partially restores cytokine mediated defective insulin secretion in INS-1E beta cells. a** INS-1E cells were cultured with IFNγ (100U/ml) plus IL-1β (5U/ml) in the presence or absence of YW3-56 for 48 hrs, as indicated. Cell lysates were subjected to electrophoresis and followed by immunoblot with anti-modified citrulline. **b** As above, INS-1E cells were cultured with IFNγ plus IL-1β for 48 hrs (black and grey bar). INS-1E cells were stimulated with glucose, KCl or pyruvate for one hr in the absence (white and black bar) or presence of YW3-56 (grey and striped bar) followed by insulin measurement (described in Methods). Data are representative of three independent experiments and the error bars indicated SD of mean. Statistical analysis was performed using Student's *t*-test.

rhGK). However, the Km of PAD-treated rhGK was two-fold changed ($16.2 \pm 2.3$ mM) compared to native rhGK ($8.48 \pm 2.1$ mM) (Fig. 5c). These data reveal that glucokinase is a substrate of PAD and suggests citrullination can modulate its glucose threshold for insulin secretion in beta cells, mediated by reduced substrate binding affinity of glucokinase enzyme.

## Discussion

The present study illustrates two unique roles of post-translational protein modifications in T1D. The first role, in compromising both B and T cell immune tolerance has now been illustrated for human glucokinase. In addition, as a biomarker of autoimmunity, inflammation-induced citrullination of GK alters its metabolic functions of glucose sensing in pancreatic beta cells. In the current study, our results reveal that citrullination alters the enzyme kinetics of glucokinase and reduces glucose stimulated insulin secretion (GSIS), the major biological function of pancreatic beta cells.

The role of protein citrullination by PAD2/PAD4 in promoting loss of self-tolerance is well studied in RA. This represents a potentially effective therapeutic axis, in that several potent PAD4 specific inhibitors have been developed and were shown to disrupt mouse and human NET formation (NETosis)[21,32–34]. NETosis has also been shown to be a major source of autoantigens in RA[33,35] and its activity is also believed to form neo-epitopes in other autoimmune disease settings[8,26,36]. We show that one pan-PAD inhibitor, YW3-56, can protect from inflammation-induced citrullination and partially restore insulin secretion in beta cells

upon stimulation with glucose or pyruvate. Besides YW3-56, a specific PAD2 inhibitor, CAY10723 (AFM30a)[37], was also found to be able to restore cytokine-suppressed insulin secretion in INS-1E cells (pyruvate stimulated insulin concentration: $65.0 \pm 2.9$ ng/ml for control, $21.6 \pm 6.5$ ng/ml for cytokines treatment and $38.7 \pm 9.5$ ng/ml for cytokines plus CAY10723 treatment). However, neither YW3-56 nor CAY10723 can completely restore cytokine-suppressed insulin secretion response to glucose or pyruvate. These observations suggest that inflammation-induced citrullination may contribute to defective glucose sensing but not be the only mechanism for defective insulin biosynthesis in T1D.

Each PAD isozyme has specific tissue distribution, functions and substrate under physiological conditions[9]. For examples, PAD2 and PAD4 are the only PAD isozymes expressed in immune cells. Among five PAD isozymes, PAD2 has the highest expression level, both of mRNA and protein, in C57Bl/6, non-obese diabetes resistance (NOR) and NOD islets. Moreover, NOD islets have higher PAD2 expression and activity compared to C57Bl/6 and NOR islets[21,38,39]. However, there is no detectable mRNA expression of Padi1, 2, 3 and 4 in rat liver[40]. Similarity, there is no Padi2 mRNA expression in C57Bl/6, non-obese diabetes resistance (NOR) and NOD liver[38]. The differences of PAD isozymes expression between liver and pancreas support why citrullinated GK is found as tissue-specific autoantigen in NOD pancreas while GK expression is much higher in NOD liver compared to pancreas.

As mentioned above, ATP-dependent GK converts glucose to glucose-6-phosphate, the critical step in glucose metabolism.

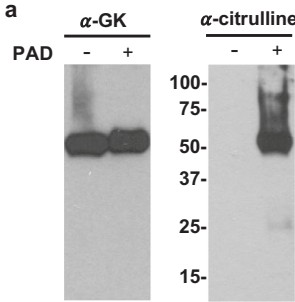

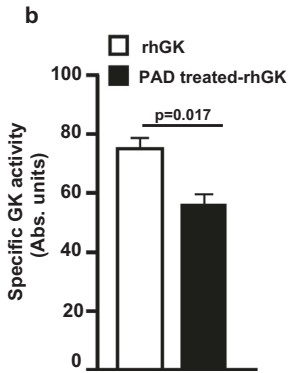

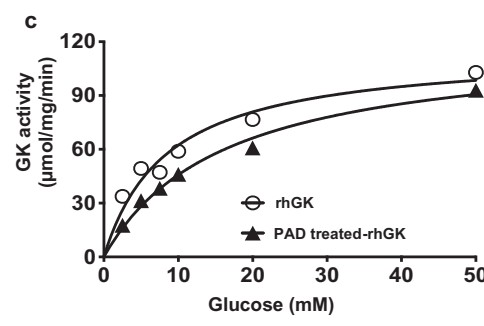

**Fig. 5 Citrullination modifications in human pancreatic glucokinase. a** The citrullinated-rhGK was generated as described in Methods based on in vitro citrullination by PAD. The native and PAD treated-rhGK were subjected to electrophoresis and followed by immunoblot with anti-glucokinase (α-GK) and anti-citrulline (α-citrulline). **b** The catalytic activity was measured spectrophotometrically at 25 mM glucose on 1 μg of rhGK and PAD treated-rhGK, respectively (see Methods). The error bars indicated SD of mean. **c** Enzyme kinetic analysis of GK was measured with seven dilutions of glucose (0–50 mM) and analyzed using GraphPad Prism Software. Km and Vmax were calculated by rhGK and PAD-treated rhGK curves, respectively, with $p$ value of 0.0093 (computed by the F-test) between two Michaelis–Menten equations.

Alternative splicing of the GK gene results in three tissue-specific isoforms, isoform 1 (pancreas) and isoforms 2 and 3 (liver). The only difference among three GK isoforms is in the amino acid sequence of exon 1 in the first fifteen amino acids of human pancreatic glucokinase[41]. Unlike other hexokinases, GK is not regulated by feedback inhibition by glucose-6-phosphate. Therefore, GK can constitutively trigger insulin secretion in pancreatic β-cells under high glucose conditions. To date, few studies demonstrate any role of PTMs in altering GK activity. Poly-ubiquitination of human GK, both pancreatic isoform 1 and hepatic isoform 2 increases catalytic activity up to 1.4 fold[42]. SUMOylation (small ubiquitin-like modifiers) of GK was found in MIN6 and INS-1 cells and results in increased pancreatic glucokinase stability and activity[43]. Recently, a randomized, double-blind and placebo-controlled study reported that one glucokinase activator, TTP399, can lower HbA$_{1C}$ and reduce hypoglycemia without increasing the risk of ketosis in patients with T1D[44]. In the current study, our data suggest that citrullination impairs the major glucose sensing function of GK in pancreatic beta cells due to the suppression of substrate binding affinity of recombinant human pancreatic GK.

Pancreatic beta cell-specific glucokinase knockout mice die within the first week of birth of severe hyperglycemia[45]. Besides NOD mice, the streptozotocin (STZ)-induced model of murine diabetes also resembles human T1D, recapitulating the phenomena of insulitis and insulin deficiency. Relevant to this study, STZ-induced diabetic mice exhibit decreased glucokinase expression with hyperglycemia[46]. Pyruvate kinase M2 (PKM2; one of four pyruvate kinase isozymes termed L, R, M1 and M2) activation protects against diabetes by increasing glucose

metabolic flux and inducing mitochondria biogenesis[47]. Of note, the catalytic activity of PK is increased 2 to 3-fold after in vitro citrullination by PAD[48]. Collectively, our data suggest that citrullinated GK and perhaps other downstream pathway proteins (indicated by Fig. 4b; pyruvate stimulated insulin secretion data) contributes to the defects of insulin secretion in beta cells.

In the current study, we found that the total citrullination levels were higher in 16 week old NOD inflamed pancreas compared to age-matched B6 pancreas. We found that both anti-GK and anti-citrullinated GK IgG are significantly higher in NOD serum compared to age-matched B10.BR serum even before the diabetic onset, as early as 4-6 weeks of age. Similarly, both anti-GK and anti-citrullinated GK IgG titers were significantly higher in patients with T1D compared to healthy subjects. Interestingly, diagnostic anti-IA2 antibodies correlated with anti-citrullinated GK in patients with T1D, raising the possibility of linked development.

The loss of immune tolerance to PTMs within the stressed beta cells triggers an autoreactive T cell response and contributes to the destruction of insulin-producing beta cells in autoimmune diabetes. Emerging data demonstrate that autoreactive T cells against variety of PTMs-neoepitopes are found in patients with T1D including oxidation, deamidation, phosphorylation and citrullination[7]. For example, CD4 T cells from patients with T1D specifically recognize the oxidized insulin neoepitopes, near disulfide bond region in A chain, in the context of HLA-DR4[49]. Previous studies demonstrated that the citrullination and transglutamination can enhance GAD65 peptide binding to *HLA-DRB1\*04:01 (DR0401)*[26]. To date, GAD65, GRP78, islet-specific glucose 6 phosphatase catalytic subunit-related protein (IGRP) and islet amyloid polypeptide (IAPP) are identified

as citrullinated-neoepitopes recognized by autoreactive T cells in the development of T1D[1]. Herein, we investigated the relevance of T cell responses to glucokinase in human disease by identifying citrullinated and unmodified peptides that are bound and presented by DR0401, a part of the high-risk T1D-associated DR4/DQ8 haplotype[25], and recognized by human CD4+ T cells. We identified a total of five immunogenic DR0401-restricted GK epitopes, one of which was citrullinated at Arg358 residue of GK. Notably, among the various GK epitopes identified through our study, the citrullinated GK 346 epitope emerged as the specificity with the highest median frequency and best differentiated controls from subjects with T1D. Consistent with this data, we observed that Arg358 residue of GK is highly susceptible to citrullination in PAD-treated recombinant human pancreatic GK protein. Moreover, Arg358 residue of GK was identified to be citrullinated in cytokine treated- but not in untreated INS-1E beta cells (Supplementary Fig. 7a, b, respectively).

Neoepitopes derived from citrullination modification of islet proteins, citrullinated GK and/or other unknown glucose-insulin metabolic proteins, compromises immune tolerance to trigger T and B cell autoimmunity. Perhaps more importantly, citrullination of GK impairs islet responses to glucose and overall glucose homeostasis. The pathway of cytokine stress and citrullination is an amplifying feedback loop of T1D pathology that includes both autoimmunity and altered beta cell metabolism. The pathway can potentially be interrupted by therapeutic manipulation of the tissue microenvironment preventing the development of citrullination (PAD inhibitors) or by the inhibition of proinflammatory cytokines in the pancreas.

## Methods

**Mice.** Female NOD/ShiLt, NOR/LtJ, C57BL/6, and B10.BR mice were purchased at three weeks of age from Jackson Laboratories (Bar Harbor, Maine). All animal studies were performed in accordance with the guidelines of the Yale University Institutional Animal Care and Use Committee. All mice were housed in micro-isolator cages with free access to food and water and maintained under the following facility conditions: temperature: 72 °C ± 2 °C; humidity: 50% ± 30%; 12:12 light/dark cycle (i.e., 12 h of light from 7 a.m.–7 p.m. then 12 h of dark from 7 p.m.–7 a.m.). Glucose levels were measured in blood withdrawn from the retro-orbital sinus of anesthetized mice using Lite glucometer and test strips (Abbott Laboratories, Abbott Park, IL, USA). Mice with blood glucose values greater than 250 mg/dL were considered diabetic. Serial serum samples were acquired from NOD and age matched control animals. Pancreas and liver tissue extract were prepared in RIPA buffer containing a mixture of protease inhibitors (Complete protease inhibitor, Roche Diagnostics).

**Protein citrullination detection.** Presence of citrullinated protein was detected by using anti-citrulline (Abcam, ab6464) for immunohistochemistry as described elsewhere[50], by using anti-peptidyl-citrulline, clone F95 (Millipore, MABN328) for flow cytometry and confocal microscopy and by using anti-modified citrulline detection kit, clone C4 (Millipore, 17-347B) for immunoblot according to manufacturer's protocol. The specificity and sensitivity of the above commercial antibodies for the detection of protein citrullination were described previously[47,48] and in our recent reports[9,51,52].

For flow cytometry, beta cells were dissociated into single-cell suspension by trypsin (Millipore) and fixed by 95% methanol and 4% paraformaldehyde for 30 min on ice. Cells were stained with anti-peptidyl-citrulline, clone F95 (Millipore), and anti-mouse Alexa Fluor 488 (Life Technology) for intracellular citrullination level and/or with anti-glucokinase (Proteintech, 19666-1-AP, 1:100 dilution) and anti-rabbit Alexa Fluor 647 (Life Technology) for glucokinase expression in staining buffer (PBS containing 1% BSA and 0.1% Tween-20). Stained cells were analyzed by FACSCalibur (BD Biosciences) with FlowJo software (Tree Star). Cell were first gated by FSC/SSC to exclude dead cells and debris and then positive and negative threshold were set using an isotype Ig control with the same fluorophore.

For confocal and fluorescence microscopy, beta cells were fixed by 95% methanol for 30 min, washed twice with PBS and permeabilized with 0.3% Triton X-100 for 2 h. Then the cells were stained with anti-peptidyl-citrulline, clone F95 (Millipore). INS-1 cells and sorted human beta cells were then observed on a Leica SP5 II laser scanning confocal microscope and fluorescence microscopy, respectively.

**Beta cell cultures and insulin secretion assay.** The INS-1 rat insulinoma cells were cultured at 37 °C in RPMI-1640 (Thermal Fisher Scientific) containing 11.2 mM glucose and 2 mM L-glutamine and supplemented with 10% fetal bovine serum (FBS), 50 μM β-mercaptoethanol, 100 units/ml penicillin, and 100 μg/ml streptomycin. For cytokine stress, the INS-1 cells were treated with recombinant mouse IFNγ (1000 units/mL; R&D Systems) and recombinant human IL-1β (50 units/mL; R&D Systems) and harvest for citrullination detection as described above.

For insulin secretion assay, the INS-1E cells, a gift from Prof. Wollheim (CMU, Geneva, Switzerland)[53], were seeded on 6 well plate and cultured with recombinant mouse IFNγ (100 units/mL; R&D Systems) and recombinant human IL-1β (5 units/mL; R&D Systems) in the presence or absence of PAD2/PAD4 inhibitor (5 μM YW3-56; Cayman) for 48 hrs. Cells were washed and starved for 1 hr in low-glucose KREBS buffer (138 mM NaCl, 4.7 mM KCl, 1 mM CaCl2, 1.18 mM KH2PO4, 1.18 mM MgSO4, 5 mM NaHCO3, 25 mM Hepes (pH7.4), 0.1% BSA and 2.5 mM glucose). Assay buffer was then replaced with KREBS buffer containing either varying glucose concentration (2.5, 5, 9 and 16.7 mM), 10 mM sodium pyruvate or 30 mM KCl. After another 1 hr incubation, supernatants were collected and assayed for insulin by using Rat High Range Insulin ELISA kit (ALPCP) according to manufacturer's protocol and analyzed using Microsoft excel Software. All experiments were performed using INS-1E cells between the 61st and 65th passage.

**Human islet cultures and cytokine treatment.** Human islets were obtained from adult, non-diabetic organ donors from Prodo Laboratories (Aliso Viejo, CA). Islets were cultured in CMRL1066 culture medium (Gibco) supplemented with 10% fetal bovine serum (Sigma), 10 mM Hepes (AmericanBio), 2mM L-glutamine (Sigma), and 1% Pen-Strep (Gibco). To modeling pancreas inflammation, human islets were treated with 25 ng/mL IFNγ (R&D) and/or 10 ng/mL TNFα (R&D) for 48 h and then dissociated into single cell suspensions using 0.05% trypsin-EDTA (Gibco). Cells were stained with FluoZin-3 (Invitrogen) and TMRE (Life Technologies) and sorted using a FACS Aria II (BD) and analyzed with FACSDiva (BD) and FlowJo (Tree Star Inc). In brief, pancreatic islet cells were first gated on SSC-A/FSC-H. Second, cells were gated on Zinc+ (ie FITC channel) cells. Third, cells were passed to a FSC-W/FSC-H and subsequently SSC-W/SSC-H to exclude doublets. Cells were then collected by zinc+ and TMRE+ cells to enrich viable beta cells.

**In vitro citrullination and identification of citrullination sites by LC-MS/MS.** For in vitro citrullination, 0.1U of rabbit peptidylarginine deiminase (PADI2; Sigma–Aldrich) were added to 50 μg of recombinant human pancreatic glucokinase (rhGK; LS-G3456, LSBio) in reaction buffer (0.1 M Tris-HCl, 10 mM CaCl2 and 5 mM DTT) at 37 °C for 2.5 hrs and kept at -80 °C in multiple aliquots to avoid freeze and thaw until used. Minimum 6 μg of rhGK and PAD-treated-rhGK were reduced, alkylated and tryptic digested for LS-MS/MS analysis. Data were collected on a QE-Plus mass spectrometer coupled to a NanoACQUITY UPLC (Waters Inc.). A Waters Symmetry® C18 180 μm × 20 mm trap column and a 1.7-μm, 75 μm × 250 mm nanoACQUITY UPLC column (35 °C) was utilized for the separation. Collected LC–MS/MS data were processed using Proteome Discoverer Software (v. 2.2, Thermo Fisher Scientific) and searched were carried out in MASCOT Search Engine (Matrix Science) using the below parameters; oxidation (M), deamidated (N/Q) and citrullination (R) as the variable modifications and carbamidomethylation (C) as a fixed modification. Mascot results were loaded into Scaffold Q + S 4.11.0 and then Scaffold PTM 3.3.0 was used to re-analyze MS/MS spectra identified as citrulline modified peptides. Calculated Ascore values for citrullinated peptides containing the neutral loss of isocyanic acid (−43 Da)[54] indicate the level of site localization probabilities to assess the level of confidence in each citrullination PTM localization. Scaffold PTM then combines localization probabilities for all peptides containing each identified PTM site to obtain the best estimated probability that a PTM is present at that particular site. Manual MS/MS inspection of the modified peptides were also carried out to ensure correct assignments/localizations of the citrullination sites (detailed MS/MS spectra are presented in Source Data-File 2 spreadsheet).

For identification of citrullination residues in intracellular glucokinase in INS-1E cells, cells were scraped from cytokine treated and control cells and centrifuged for 5 min at 170 x g. The cell pellet was lysed in 400 μL 1% CHAPS buffer (1% CHAPS, 100 mM KCl, 150 mM NaCl, 50 mM Tris-HCl pH 7.5 and a mixture of protease inhibitors (Complete protease inhibitor, Roche Diagnostics)) and kept on ice for 30 min with occasional tapping. Cell debris was removed by centrifuging 10 min at 4 °C at 17,000 x g. Proteins smaller than 30 kDa were removed by passing the protein lysate through an Amicon Ultra – 15 Centrifugal Filter (Millipore). One mg of cell lysate in 500 μL lysis buffer was pre-cleared by adding 10 μL of protein A/G PLUS-agarose beads (Santa Cruz) and incubating for 1 h at 4 °C on a rotator. Cell lysate was centrifuged for 4 min at 4 °C at 400 × g and the supernatant was incubated overnight at 4 °C on a rotator with 5 μg of anti-glucokinase antibody (Proteintech, 19666-1-AP, 1:50 dilution). Thirty μL of protein A/G PLUS-agarose beads were washed with 500 μL wash buffer (500 mM NaCl, 50 mM Tris-HCl pH 7.5 and 0.05% (v/v) Tween-20) and with 200 μL 1% CHAPS buffer. The cell lysate was added to the beads and incubated overnight at 4 °C on a rotator. The samples were washed 4 times with 250 μL 1% CHAPS buffer and 3 times with pre-urea buffer (50 mM Tris pH 8.5, 1 mM EGTA and 75 mM KCl). Then 75 μL urea elution

buffer (7 M urea, 20 mM Tris pH 7.5 and 100 mM NaCl) was added and the samples were rotated for 30 min at room temperature. The beads were pelleted and the supernatant was kept aside. This process was repeated twice to ensure that all the captured proteins are released from the beads. Eluted proteins were protein precipitated using the Wessel–Flügge method[55]. The protein precipitates were dissolved in urea buffer (7 M urea in 1 M Tris-HCl pH 8). Samples were prepared for analysis on LC–MS/MS as above and run on a QExactive Orbitrap (Thermo Fisher Scientific). Peptides were identified by MASCOT (Matrix Science) using SwissProt (Homo sapiens, 169779 entries) as a database via Proteome Discoverer 2.2, incorporating Percolator for peptide validation. Same parameters of variable modifications and fixed modification were set as mentioned above. Two missed cleavages were allowed, peptide tolerance was set at 5 ppm and MS/MS tolerance at 20 mmu. The MS/MS spectra were carefully checked manually for the presence of citrullinated residues.

**ELISA and immunoblot assay**. Immunoreactivity of human and mouse serum to rhGK and PAD-treated-rhGK was performed by ELISA. Briefly, 0.5 μg of rhGK or PAD-treated-rhGK in 0.05 M carbonate-bicarbonate buffer (pH = 9.6; Sigma) was coated onto ELISA plates (Thermo Fisher Scientific) overnight at 4 °C and blocked with 1% BSA in PBST containing 0.05% Tween-20. Sera were diluted as 1:100 in diluting buffer (0.3% BSA in PBST) and incubated 2 h at room temperature. Species-specific goat anti-IgG alkaline phosphatase was used as a secondary reagent (Southern Biotech). Color was developed via the addition of pNPP substrate (Sigma) and absorbance was read at 405 nm (Synergy HT Multi-Mode Reader, BioTek Instruments). Individual signals were normalized to no-antigen control wells. Rabbit polyclonal antibody against glucokinase (Proteintech, 19666-1-AP, 1:1000 dilution) served as positive control in ELISA. All readings were normalized to non-specific serum binding to no-antigen control wells. Autoantibodies against to rhGK or PAD-treated-rhGK was designating as positive with an OD >2 standard deviation (SD) above B10.BR serum or human healthy serum.

For immunoblotting, 1 μg of rhGK or PAD-treated-rhGK was subjected to electrophoresis, blotted onto a nitrocellulose membrane, and probed with serum (1:100) from B10.BR or NOD mice and incubated with the alkaline phosphatase-conjugated anti-mouse IgG, then visualized by NBT/BCIP substrate (Thermo Fisher Scientific).

**Catalytic activity of recombinant human pancreatic glucokinase**. After in vitro citrullination by PAD as described above, recombinant human pancreatic glucokinase (rhGK; LS-G5486, LSBio) was put on the ice to stop the reaction for 5 min. Then glucokinase activity was measured spectrophotometrically (A340nm) at 37 °C by a glucose 6-phosphate dehydrogenase (G6PDH) coupled assay as described previously[56]. In brief, 1 μg rhGK or PAD-treated rhGK was assayed in a reaction buffer containing 25 mM Hepes (pH7.3), 2 mM MgCl2, 1 mM DTT, 0.5 mM NADP, 2 mM ATP, 0.01%BSA, 20U/ml G6PDH and 25 mM D-( + )-glucose. Relative specific activity of glucokinase was calculated from linear regression of the change in A340nm. To determine the steady-state kinetic properties of both rhGK and PAD-treated rhGK, varying concentrations of D-( + )-glucose (0–50 mM) were used. Vmax and Km were calculated by Michaelis-Menten equation using GraphPad Prism Software.

**Human serum and peripheral blood samples**. Two sets of human samples were used in this study. For anti-glucokinase and anti-citrullinated glucokinase antibodies by ELISA, human serum samples were collected from subjects diagnosed with T1D (n = 55; 29 women, median age 13 years, range 5–76 years; 0.1–55 years of disease duration) and healthy control subjects (n = 18; 9 women, median age 29.5 years, range 20–59 years). Samples were assayed in a blinded fashion for ELISA as described above. The presence or absence of autoantibodies against to GAD65, IA2, ZnT8 and insulin were determined by Barbara Davis Center Autoantibody/HLA Service Center using standardized radioimmunoassay.

For HLA tetramer staining assays, peripheral blood and serum were collected from 10 individuals with T1D and 10 healthy controls with DRB1*04:01 haplotypes after obtaining written consent under a study approved by the Institutional Review Board at the Benaroya Research Institute. Subject attributes are summarized in Supplementary Tables 4 and 5. IRB approval for both sample sets was sought and received from the Benaroya Research Institute IRB (Protocols IRB07109 and IRB10024); written informed consent was obtained from all study participants.

**Peptide prediction**. The probability that citrullinated and unmodified glucokinase peptides would be bound and presented by DR0401 was evaluated based on a previously published prediction matrix[6,26]. Briefly, coefficients corresponding to each anchor residue for all possible core 9-mers within the protein were multiplied, yielding a predicted relative binding affinity score. Sequences were chosen to include at least 2 flanking residues on each side of the predicted minimal 9-mer motif.

**Peptide binding measurements**. Peptides with predicted binding to DR0401 (Supplementary Table 3) were synthesized (Sigma) and their binding to DR0401

was assessed through a competition assay, as previously described[57]. Briefly, increasing concentrations of each citrullinated glucokinase peptide were incubated in competition with a biotinylated reference influenza hemagglutinin peptide (HA306-318) at 0.02 μM in wells coated with DR0401 protein. After washing, residual biotin-HA306-318 was detected using europium-conjugated streptavidin (Perkin Elmer) and quantified using a Victor2D time resolved fluorometer (Perkin Elmer). Curves were simulated using Prism software (Version 5.03, GraphPad Software Inc.) and IC50 values calculated as the concentration needed to displace 50% of the reference peptide.

**HLA class II protein and tetramer reagents**. DR0401 protein was purified from insect cell cultures as previously described[58,59]. Monomers were loaded with 0.2 mg/ml peptide at 37 °C for 72 h in the presence of 2 mg/ml n-Dodecyl-β-maltoside and 1 mM Pefabloc (Sigma–Aldrich). Peptide-loaded monomers were conjugated into tetramers using R-PE streptavidin (Invitrogen), PE-Cy5 streptavidin (BD), or PE-CF594 streptavidin at a molar ratio of 8:1.

**In vitro tetramer assays and T cell clone isolation**. Peripheral blood mononuclear cells (PBMCs) were isolated by Ficoll underlay, resuspended in T cell media (RPMI, 10% pooled human serum, 1% Penicillin–Streptomycin, 1% L-glutamine) at 4 × 10^6 cells/mL, and stimulated with peptides (20 μg/mL total) in 48-well plates for 14 days, adding medium and IL-2 starting on day 7. Cells were stained with individual tetramers for 75 min at 37 °C, followed by CD4 BV650 (BD Biosciences), CD3 APC (eBioscience), and CD25 FITC (BioLegend) for 15 min at 4 °C, run on a FACSCanto (BD), and analyzed using FlowJo (Tree Star Inc). Clones were isolated by sorting single tetramer-positive CD4+ T cells using a FACS Aria (BD) and expanded in 96-well plates in the presence of 1 × 10^5 irradiated PBMC, 2 μg/mL phytohaemagglutinin (Remel Inc.), adding media and IL-2 starting on day 10.

**T cell clone maintenance and characterization**. Clones specific for citrullinated glucokinase peptides were maintained in supplemented RPMI and re-stimulated using PHA (2 μg/mL; Remel) every two weeks. Specificity was confirmed by re-staining expanded clones with tetramer.

**Ex vivo tetramer analysis**. Analysis of T cell frequency was accomplished using our previously published approach[28]. Briefly, 40–60 × 10^6 PBMCs were resuspended in a total of 400–600 μL of media, divided into two independent tubes of 20 × 10^6 cells (200 μL) each, incubated with 50 nM dasatinib for 10 min at 37 °C, and stained with 20 μg/mL of PE-labeled, PE-CF594-labeled, or PE-Cy5-labled tetramers at room temperature for 120 min (three tetramers per tube, a total of six glucokinase tetramers). Cells were washed, incubated with PE-magnetic beads (Miltenyi) for 20 min at 4 °C and magnetically enriched, retaining 1% of the cells as a non-enriched sample. Enriched (bound) and non-enriched (pre-column) samples were stained with CD4 V500, CD14 PerCP-Cy5.5, CD19 PerCP-Cy5.5 (eBioscience), CD45RA AF700 (BD), CXCR3 FITC, CCR6 BV421, and CCR4 BV605 (BioLegend) for 15 min at 4 °C. After washing, cells were labeled with ViaProbe (BD Biosciences) and analyzed on a FACSLSRII (BD Biosciences), gating on CD4+CD14−CD19−ViaProbe− cells and plotting tetramer versus CD45RA. In brief, cells were first gated on FSC-A/SSC-A to determine bulk lymphoctye population. Second, cells were passed to a FSC-W/FSC-H gate to exclude FSC-doublet cells. Third, cells were passed to a SSC-W/SSC-H gate to exclude SSC-doublet cells. Cells were then gated on Viability/CD4. Viability- CD4+ cells were passed on and used to determine individual's memory (CCR7/CD45RA) and chemokine (CXCR3/CCR4/CCR6) expression, which was used later when applied to individual person's tetramer+ cells. Higher order positives were removed from each tetramer-channel by plotting corresponding other tetramer channels vs each other and excluding all higher order positives. Frequencies were calculated as previously described[28].

**Statistical analysis**. Statistics were performed using a Student's unpaired two-tailed t test unless indicated. A value of p < 0.05 was regarded as significant.

**Reporting summary**. Further information on research design is available in the Nature Research Reporting Summary linked to this article.

## Data availability

The mass spectrometry proteomics data have been deposited to the ProteomeXchange Consortium via the PRIDE partner repository: http://www.ebi.ac.uk/pride/archive/projects/PXD028825. All other data supporting the findings of this study are available within the article, supplementary file and Source Data files. Source data are provided with this paper.

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

## Acknowledgements

We thank members of the Mamula laboratory, Emily Schroeder and Tyler Masters for excellent technical assistant for this study. This research was supported by the Juvenile Diabetes Research Foundation (Innovative Grant to M.J.M., 2-SRA-2018-551-S-B to E.A.J., and 3-SRA-2019-791-S-B to C.S.), and National Institutes of Health (DK104205-01 to K.H. and M.J.M.). We also thank the MS & Proteomics Resource at Yale University for providing the necessary mass spectrometers and the accompany biotechnology tools funded in part by the Yale School of Medicine and by the Office of The Director, National Institutes of Health (S10OD02365101A1, S10OD019967, and S10OD018034). The funders had no role in study design, data collection and analysis, decision to publish, or preparation of the manuscript.

## Author contributions

M.L.Y. designed and performed the experiments, analyzed the data, and wrote the manuscript. S.H., R.G., P.G., A.L.P., and A.C. performed the experiments and analyzed data; T.T.L. and J. K. performed the proteomic PTM analysis and edited the manuscript. C.S. and C.J.G. were responsible for subject selection and clinical characterization and edited the manuscript. E.A.J. and L.O. designed the experiments, summarized data and co-wrote the manuscript. R.G.K., and K.C.L. provided helpful advice and edited the manuscript. M.J.M. directed the project, edited the manuscript and was responsible for coordination and strategy.

## Competing interests

The authors declare no competing interests.
