## [Peer Review File · Nature Communications]

Citrullination of glucokinase is linked to autoimmune diabetesREVIEWER COMMENTS

Reviewer #2 (Remarks to the Author):

Yang et al present evidence that CD4+ T cell and B cell response to citrullinated glucokinase develop in type 1 diabetes and suggest that this modification attenuates glucokinases' function. Overall, this is a clearly written and interesting manuscript. However, there are some issues that require addressing. These are listed below.

Major issues:

Is the cytokine-induced protein citrullination specific for the beta cell line? What happens if a liver cell line, for example, is treated in the same way?

Does the cocktail of cytokines promote citrullination in NOD islets in vitro? Or, even better, human islets? This would be much more relevant than INS-1 cells.

What has been done to validate the specificity of the anti-citrulline antibodies used for staining? Can the staining be blocked by citrulline containing peptide, but not by the same peptide with arginine instead of citrulline?

The relevance of the antibody data is not clear to me. The authors present raw OD values and show that there is a distribution of OD values for conditions, but when viewed as a population there are weak statistically significant differences. How do small difference in OD (~0.05) relate to the titer of antibody? Given the vagaries of an ELISA similar variability in OD readings could simply be technical in nature and not attributable to difference in the concentration of antibodies (notwithstanding the statistical difference reported). The authors need to exclude this possibility.

I do not think B10.Br mice are the appropriate control strain. NOR mice, which are much more closely related to NOD would be a better control. The data presented does not excluded the possibility that NOD mice have more anti-citrulline antibody independently of autoimmune diabetes.

Minor points

The title should read "Citrullination of glucokinase is linked to autoimmune diabetes"

Line 162: "Citrulline didn't not enhance or alter binding (to HLA?) of any peptide, indicating that the citrulline/Arg are not T-cell contact residues". This statement is unclear to me. I assume the authors mean that if the presence of citrulline enhanced the binding affinity to HLA this would indicate that this residue filled an HLA binding pocket, since it does not, this leaves them to be TCR contact residues. This is a 'long-bow' to draw from this data and needs to be re-worded to make the reasoning clear to the reader.

Line 166: Individual peptides were labelled with PE, PE-CF594 or PE-labelled tetramers. This is also unclear. I assume the peptides weren't labeled directly but use to make peptide/HLA tetramers which were labelled as indicated. Please reword to make this clear.

Line 177: the word 'trended' is repeated unnecessarily. Please delete.

Response to Reviewers:

NCOMMS-21-23349-T “Citrullination of glucokinase linked to autoimmune diabetes”

We appreciate the Reviewers’ supportive feedback and constructive suggestions. We carefully addressed all the comments and provided data from additional experiments in the revision as detailed in a point-by-point response below (text revisions indicated by red font).

In addition, mass spectrometry proteomics data have been deposited to the ProteomeXchange Consortium via the PRIDE [1] partner repository with the dataset identifier PXD028825. The dataset can be accessed with Reviewer account as:

Username: reviewer_pxd028825@ebi.ac.uk

Password: AhpqjRn2

We have previously replied to reviewers #2 (maintained in the text below). We have inserted specific replies to reviewer 1 comments above those prior comments to reviewer #2.

REVIEWER COMMENTS

Reviewer #1 (Remarks to the Author):

Yang et al (2021) Citrullination of glucokinase linked to autoimmune diabetes.

The manuscript by Yang et al describes their findings that the citrullination of glucokinase, the liver specific form of hexokinase, occurs in Type 1 diabetes. Specifically, they identified autoantibodies to both glucokinase and citrullinated glucokinase in patients with Type 1 diabetes and NOD mice. They also show that citrullination increases the Km of glucokinase by 2-fold. Since glucokinase acts as a sensor of glucose levels and glucose concentrations normally operates around the Km of glucokinase, this seemingly small change is nonetheless significant. The authors also identify multiple sites of citrullination in glucokinase both using recombinant protein and cytokine stimulated cells. I expect that the manuscript will be widely appreciated by those in the field and beyond. As such, this reviewer recommends publication once the authors address the following issues.

We thank the reviewer’s encouraging evaluation of our work and highlighting its value within the scientific community.

1. The authors should provide a stronger rationale for why they chose to focus on glucokinase versus all the other proteins that are citrullinated in the pancreas.

Glucokinase is the first rate-limiting step of glycolysis and serves as a glucose sensor to initiate glycolysis and insulin signaling in pancreas to maintain glucose homeostasis. The manuscript emphasizes the role of glucokinase mutation in patients with MODY and ongoing clinical trials of glucokinase activator in T2D as mentioned in the manuscript. Very recently (April 2021), Klara Klein (University of North Carolina School of Medicine, Chapel Hill, USA) and colleagues reported one glucokinase activator, TTP399, that can improve glycemic control in

patients with T1D¹. We have added this promising finding in the Discussion: “Recently, a randomized, double-blind and placebo-controlled study reported that one glucokinase activator, TTP399, can lower HbA_{1C} and reduce hypoglycemia without increasing the risk of ketosis in patients with T1D.”

By using pyruvate to bypass glucose to stimulate insulin secretion, we found one pan-PAD inhibitor, YW3-56, was also able to correct cytokine-mediated suppression of insulin secretion (Fig. 4b). It hints that inflammation triggered citrullination affects not only glucokinase but also downstream metabolism of glycolysis in the insulin biosynthesis pathway. In addition, our pilot study identified that one isozyme of pyruvate kinase, PKM2, was citrullinated in IFN- γ and TNF- α -treated human islets but not in untreated group. PKM2 catalyzes a downstream rate-limiting step in the glycolytic pathway. Ongoing studies of citrullinated PKM2 in beta cells is a topic of active investigation in the Mamula and Kibbey laboratories.

2. YW3-56 should be classified as a pan-PAD inhibitor.

We agreed with the reviewer and have added “pan-PAD inhibitor” and IC₅₀ to make clear the classification of YW3-56 in the Results to: “We also utilized YW3-56, a pan-PAD inhibitor (IC₅₀: 5.5-1 μ M and 1-5 μ M for PAD2 and PAD4, respectively), in studies of GSIS...”.

3. CAY10723 should be referred to by its original name, i.e. AFM30a [J Med Chem, 30, 3198].

Thank you for the comment and we edited the sentence in the Discussion to: “Besides YW3-56, a specific PAD2 inhibitor, CAY10723 (AFM30a)², was also found to be able to.....” and also have added the manuscript reviewer referred to the References.

4. Are the differences in Km between glucokinase and citrullinated glucokinase statistically significant?

The enzyme kinetic analysis of rhGK and PAD-treated rhGK was measured with different substrate concentrations (0-50mM of glucose) and analyzed using GraphPad Prism Software. Km and Vmax were calculated by rhGK and PAD-treated rhGK curves, respectively, with p value of 0.0093 (computed by the F-test) between two Michaelis-Menten equations. We add the analysis statement in the figure legend of Fig. 5d.

5. The authors should supply the sequences of all citrullinated peptides identified along with MS/MS spectra. The authors should also annotate the neutral loss of isocyanic acid (-43 Da) which are diagnostic for the detection of sites of citrullination [PMID: 32005680].

Thank you for the constructive suggestion. We re-analyzed our proteomic data and manually inspected citrullination with the neutral loss of isocyanic acid feature. The updated citrullination sites in PAD-treated rhGK are indicated as bold capital “R” in revised Fig. 5b. Per reviewer’s request, we have added a supplementary Table 6 to show all citrullinated peptides identified in PAD-treated rhGK. The representative MS/MS spectra with annotation of the neutral loss of isocyanic acid for all citrullinated sites identified in PAD-treated rhGK are listed in the separate spreadsheets of “Source Data-File 2” excel file.

Based on the revised proteomic analysis, we edited the Results to “Mass spectrometry identified 11 citrullination modification residues in PAD-treated rhGK (Fig. 5b and all citrullinated peptides identified in PAD-treated rhGK are listed in supplementary Table 6). Thus, 34% of Arg residues (11 Arg out of total 32 Arg residues) were available to be citrullinated *in vitro* by PAD.”

The neutral loss of isocyanic acid (-43 Da)³ is annotated in the revised version in Methods: “Mascot results were loaded into Scaffold Q+S 4.11.0 and then Scaffold PTM 3.3.0 was used to re-analyze MS/MS spectra identified as citrulline modified peptides. Calculated Ascore values for citrullinated peptides containing the neutral loss of isocyanic acid (-43 Da)⁵⁴ indicate the level of site localization probabilities to assess the level of confidence in each citrullination PTM localization. Scaffold PTM then combines localization probabilities for all peptides containing each identified PTM site to obtain the best estimated probability that a PTM is present at that particular site. Manual MS/MS inspection of the modified peptides were also carried out to ensure correct assignments/localizations of the citrullination sites (detailed MS/MS spectra are presented in Data Source-File 2 spreadsheet).”

For the citrullination site of glucokinase in cell system, i.e. cytokines-treated and untreated INS-1E cells, the mass and retention time of the peptide are added above the individual MS/MS spectra (supplementary Fig. 6a and b). Although there is no neutral loss of isocyanic acid peak found in supplementary Fig. 6a, we would like to point out that the isocyanic acid loss peak is not the only proof of citrullinated peptide. In addition, when isocyanic acid loss peak is not present or not detectable, this does not prove the absence of citrullination. As indicated in supplementary Fig. 6a and b, the difference in mass between the citrullinated (2508.33169 Da) and native (2507.34574 Da) spectrum is 0.98595 Da, not 1 Da, which indicates we are not looking at the C13 isotopic peak (1.1% of C atoms in nature are C13). In addition, the difference in retention time is 3.42 min, with the modified peptide eluting later from the column, what is to be expected for a citrullinated peptide with loss of a positive charge. Isotopes would elute at the same time as the native peptide, so this is again an indication that we are not looking at an isotope. Moreover, diagnostic ions y_{13}^{2+} , y_{14}^{2+} , y_{15}^{2+} , y_{16}^{2+} , b^4+ and b^5+ rule out N deamidation. With the extra proof of the different retention time, mass shift and diagnostic ions, we believe R358 residue in glucokinase from cytokine treated INS-1E cells is confirmed to be citrullinated and not in untreated control INS-1E cells.

Reviewer #2 (Remarks to the Author):

Yang et al present evidence that CD4+ T cell and B cell response to citrullinated glucokinase develop in type 1 diabetes and suggest that this modification attenuates glucokinases' function. Overall, this is a clearly written and interesting manuscript. However, there are some issues that require addressing. These are listed below.

We appreciate the Reviewer's suggestions and have done additional studies and revisions to carefully address all comments below.

Major issues:

Is the cytokine-induced protein citrullination specific for the beta cell line? What happens if a

liver cell line, for example, is treated in the same way?

It is clearly important to distinguish potential cytokine-induced protein modifications between the liver and pancreas, two organs expressing glucokinase (GK). Based on our NOD mouse data (Fig. 1), multiple proteins undergo citrullination in inflamed NOD pancreas but not in NOD liver (from the same mouse). Additionally, we immunoprecipitated GK from both pancreas and liver of NOD mice. Citrullinated glucokinase was observed only in pancreas, not liver, under conditions of identical ‘inflammation’ to both organs (Figure 1). Moreover, the tissue distribution of PAD isozymes responsible for citrulline modification supports the above observations. Among five PAD isozymes, PAD2 has the highest expression level, both of mRNA and protein, in the pancreas (islets) of C57Bl/6, non-obese diabetes resistance (NOR) and NOD mice. NOD islets have higher PAD2 expression and activity compared to C57Bl/6 and NOR islets^{4,5}. However, there is no detectable mRNA expression of Padi1, 2, 3 and 4 in rat liver³. Similarly, there is no Padi2 mRNA expression in C57Bl/6, non-obese diabetes resistance (NOR) and NOD liver⁶. The differences of PAD isozyme expression between liver and pancreas support why citrullinated glucokinase is found as a tissue specific autoantigen in NOD pancreas but not in liver. These comments have been added to the manuscript, specifically, a new paragraph in the Discussion and three new references to address tissue specific citrullination in pancreas versus liver.

Does the cocktail of cytokines promote citrullination in NOD islets in vitro? Or, even better, human islets? This would be much more relevant than INS-1 cells.

Indeed, this is an important topic that we now emphasize in the revised manuscript. Per reviewer’s suggestion, we have examined citrullination by the staining of cytokine treated-human islets compared to non-cytokine treated islets. Consistent with INS-1 cells, we show that cytokines promote glucokinase citrullination in human beta cells (new data as supplementary Fig. 1e)

In support of the reviewer’s comment, we performed H&E staining for different ages of NOD pancreas in parallel with citrulline staining of islets (Figure 1b). We demonstrate that islet citrullination occurs just prior to insulinitis (3.5 week old NOD) and the islet citrullination is increased over time of disease (9 weeks and 16 weeks old), parallel with infiltration of lymphocytes into the islet (new data in the revised Figure 1b). These findings do clearly suggest that citrullination is induced by spontaneous inflammation/cytokines in NOD islets. Since NOD islets are already naturally inflamed (as indicated by immune cell infiltration), we believe that further exposing NOD islets *in vitro* to inflammatory cytokines will not best answer this question. Indeed, this is the rationale to compare the citrullination level between naturally inflamed NOD islets and control, presumably non-inflamed C56BL6 islets.

In support of confirming citrulline modification amplified in the inflamed pancreas, we examined islets from 10-week-old C57Bl/6 and NOD mice by LC-MS/MS analysis, and evaluated protein citrullination (Q Exactive Orbitrap). Spectra were analyzed manually to confirm citrullination, as described previously (in PMID 33372785). From the proteomic data, Padi2 protein expression was observed by quantitative Progenesis analysis. As summarized in the table below, the results clearly indicate a higher level of Padi2 expression in NOD vs

C57BL6 islets, paralleled by a higher number of citrullinated peptides as measured by LC-MS/MS. This further underscores the conclusion from our study, that cytokines are inducing citrullination in beta cells (as shown in INS-1 cells) and that this is also reflected *in vivo* in NOD mice through increased citrullination in inflamed islets.

Islets	C57Bl/6	NOD
Number of citrullinated peptides	0	25
Percentage of citrullinated peptides	0%	0.09%
Padi2 abundance (normalized abundance)	81006.3	672633.5
Relative Padi2 abundance	1	8.3

What has been done to validate the specificity of the anti-citrulline antibodies used for staining? Can the staining be blocked by citrulline containing peptide, but not by the same peptide with arginine instead of citrulline?

We apologize that no such information was provided in the previous submitted manuscript. We used three different commercial antibodies for the detection of protein citrullination depending on the methodology utilized. The use of the anti-citrulline detection methods are individually defined in the manuscript. Specifications of citrulline binding probes are listed below:

- 1) Anti-modified citrulline detection kit, clone C4; Millipore (17-347B): This antibody does not recognize citrulline residues directly but instead binds a chemically modified form, 2, 3-butanedione monoxime and antipyrine derivatized-citrulline. Per the commercial datasheet about quality control testing, “Signal is only detected in Histone H3 samples subjected to peptidylarginine deiminase 4 (PAD4) catalyzed citrullination.” The representative lot data is shown as below:

Lane 1: recombinant Histone 3 (rH3) with citrulline modification, without PAD4 treatment

Lane 2: PAD4 treated rH3 with citrulline modification

Lane 3: rH3 without citrulline modification

Lane 4: PAD4 treated rH3 without citrulline modification

We used this anti-modified citrulline detection kit to confirm that human recombinant glucokinase (rhGK) is a substrate of PAD2 *in vitro*. As shown in Figure 5a, PAD2 treated rhGK migrates slightly slower on SDS-PAGE due to the loss of positive charge (the left panel of Figure 5a) and the citrullination of rhGK was confirmed by using the anti-modified citrulline detection kit (the right panel of Figure 5a).

- 2) Anti-peptidyl-citrulline, clone F95; Millipore (MABN328): The first antibody directly recognizes peptidyl-citrulline, denoted F95, developed by Nicholas and colleagues^{7, 8}. Commercialized monoclonal F95 antibody has since been extensively used in different applications including tissue staining (immunohistochemistry and immunofluorescence), immunoblot and ELISA^{9, 10, 11}.
- 3) Anti-citrulline antibody; Abcam (ab6464): This polyclonal antibody directly recognizes peptidyl-citrulline but not free citrulline. Per the commercial datasheet, antibody specificity was performed by *competition ELISA* test by using conjugated citrulline glutaraldehyde protein. The specificity validation data is summarized as below table:

Compound	Cross-reactivity ratio
Citrulline-G-BSA	1
Arginine-G-BSA, Glutamate-G-BSA, Ornithine-G-BSA	1>100,000
Conjugated Citrulline and conjugated Homo-citrulline	1:1000

G-Glutaraldehyde, BSA=Bovine Serum Albumin

The specificity and sensitivity of the above commercial antibodies for the detection of protein citrullination were discussed previously^{12, 13} and in our recent review¹⁴. We have now clarified the different detection applications with specific commercial catalog numbers and added two new references about the specificity and sensitivity of these antibodies (in Methods) to the revised manuscript.

The relevance of the antibody data is not clear to me. The authors present raw OD values and show that there is a distribution of OD values for conditions, but when viewed as a population there are weak statistically significant differences. How do small difference in OD (~0.05) relate to the titer of antibody? Given the vagaries of an ELISA similar variability in OD readings could simply be technical in nature and not attributable to difference in the concentration of antibodies (notwithstanding the statistical difference reported). The authors need to exclude this possibility.

We acknowledge the distribution of OD values for the ELISA assay. However, it is important to emphasize the conditions of analysis:

- 1) The raw OD values illustrated for the murine ELISA data (Figure 2b and supplementary Figure 2) and human ELISA data (Figure 2c) were performed *on the same time/day* (identical plates), respectively and reported as representative data from at least three different ELISAs. Assaying all sera in a single assay eliminates the technical variance of ELISAs that may be found when examining some sera on one day and other sera in separate assay days. Per the reviewer's suggestion, the exact *p* values are now indicated in revised Figure 2b&c and supplementary Figure 2.
- 2) The prevalence and titer of anti-GK antibodies in T1D resembles that of other autoantibody specificities. For example, the prevalence of different autoantibodies is at a wide range among individual T1D patients^{15, 16}. For example, approximately 80% of T1D patients are reported positive for GADA. However, ICA and IA-2A autoantibodies in T1D range from 69-90% and

54-75%, respectively. In addition, up to 8% of healthy subjects older than 50 years have low titers of GADA¹⁷. These observations are similar to other autoimmune diseases, such as antinuclear antibodies (ANAs) as the gold standard for lupus diagnosis. Approximately three-quarters of lupus patients test ANA positive at least 3 years *before* the clinical confirmation of their lupus diagnosis¹⁸. In addition, from 5-30% of healthy subjects have a positive ANA as well^{19, 20, 21}.

3) Importantly, besides measuring the autoantibody against glucokinase and/or citrullinated-glucokinase in T1D patients, we also demonstrated that there was a significant positive correlation between the combined frequency of glucokinase-reactive T cells and levels of anti-glucokinase antibodies (Figure 3e).

I do not think B10.BR mice are the appropriate control strain. NOR mice, which are much more closely related to NOD would be a better control. The data presented does not excluded the possibility that NOD mice have more anti-citrulline antibody independently of autoimmune diabetes.

We agree with the reviewer's suggestion to use NOR serum as the control strain to exclude the possibility that NOD mice have more anti-citrulline antibody independently of autoimmune diabetes. As suggested, we have collected serum from different ages of NOR mice and examined anti-GK autoantibodies by ELISA. In the new data in supplementary Fig. 2c, there is no anti-GK or anti-citrullinated GK observed in either young (4-6 weeks of age) or older (14-20 weeks of age) NOR mice.

In support of our original use of mouse strains, not all murine models develop T and/or B cell immunity targeting citrullinated proteins/peptides. For example, DR4-transgenic B6 mice, but not wildtype B6 mice, developed antibodies targeting citrullinated peptides and anti-cyclic citrullinated peptide antibodies (anti-CCP2) after immunization with homocitrullinated peptide²². Of interest, T cells to citrullinated peptides were identified in B10.BR mice upon immunization with highly immunogenic hen egg-white lysozyme (HEL) protein²³. Thus, B10.BR mice were originally used in this study as control mouse compared to NOD mice for studying anti-citrullinated GK antibody.

Minor points

The title should read "Citrullination of glucokinase is linked to autoimmune diabetes"

We made the changes as suggested.

Line 162: "Citrulline didn't not enhance or alter binding (to HLA?) of any peptide, indicating that the citrulline/Arg are not T-cell contact residues". This statement is unclear to me. I assume the authors mean that if the presence of citrulline enhanced the binding affinity to HLA this would indicate that this residue filled an HLA binding pocket, since it does not, this leaves them to be TCR contact residues. This is a 'long-bow' to draw from this data and needs to be reworded to make the reasoning clear to the reader.

We agree with the reviewer comments and have modified the sentences in the Results to: “The results indicated that citrullination did not significantly alter binding for any of the peptides, demonstrating that the citrulline/Arg residues are not involved with HLA binding. Consistent with this observation, the predicted minimal motifs (underlined in Table 1) place the citrulline/Arg residues for GK266, GK270, and GK346 in positions (p5, p-1, and p10, respectively) that are known to be potential T cell contact residues.”

Line 166: Individual peptides were labelled with PE, PE-CF594 or PE-labelled tetramers. This is also unclear. I assume the peptides weren't labeled directly but use to make peptide/HLA tetramers which were labelled as indicated. Please reword to make this clear.

The reviewer is correct, thank you. We revised the text to make clear that “Individual GK peptides were used to prepare HLA class II tetramers. Tetramers were labeled with PE, PE-CF594, or PE-Cy5, staining two separate aliquots of cells to assess the five GK epitopes simultaneously (representative results shown in Supplementary Fig. 5). “

Line 177: the word 'trended' is repeated unnecessarily. Please delete.

We deleted the repeated word “trended”.

References:

1. Klein KR, *et al.* The SimpliciT1 Study: A Randomized, Double-Blind, Placebo-Controlled Phase 1b/2 Adaptive Study of TTP399, a Hepatoselective Glucokinase Activator, for Adjunctive Treatment of Type 1 Diabetes. *Diabetes Care* **44**, 960-968 (2021).
2. Muth A, *et al.* Development of a Selective Inhibitor of Protein Arginine Deiminase 2. *J Med Chem* **60**, 3198-3211 (2017).
3. Salinger AJ, *et al.* Technical comment on "Synovial fibroblast-neutrophil interactions promote pathogenic adaptive immunity in rheumatoid arthritis". *Sci Immunol* **5**, (2020).
4. Rondas D, *et al.* Citrullinated glucose-regulated protein 78 is an autoantigen in type 1 diabetes. *Diabetes* **64**, 573-586 (2015).
5. Azoury ME, *et al.* CD8+ T cells variably recognize native versus citrullinated GRP78 epitopes in type 1 diabetes. *Diabetes*, (2021).
6. Ishigami A, Asaga, H., Ohsawa, T., Akiyama, K and Maruyama, N. Peptidylarginine deiminase type I, type II, type III and type IV are expressed in rat epidermis. *Biomedical Research* **22**, 63-65 (2001).

7. Nicholas AP, Whitaker JN. Preparation of a monoclonal antibody to citrullinated epitopes: its characterization and some applications to immunohistochemistry in human brain. *Glia* **37**, 328-336 (2002).
8. Nicholas AP, *et al.* Immunohistochemical localization of citrullinated proteins in adult rat brain. *J Comp Neurol* **459**, 251-266 (2003).
9. Sohn DH, *et al.* Local Joint inflammation and histone citrullination in a murine model of the transition from preclinical autoimmunity to inflammatory arthritis. *Arthritis Rheumatol* **67**, 2877-2887 (2015).
10. Nicholas AP, Sambandam T, Echols JD, Barnum SR. Expression of citrullinated proteins in murine experimental autoimmune encephalomyelitis. *J Comp Neurol* **486**, 254-266 (2005).
11. Zhou Y, Di Pucchio T, Sims GP, Mittereder N, Mustelin T. Characterization of the Hypercitrullination Reaction in Human Neutrophils and Other Leukocytes. *Mediators Inflamm* **2015**, 236451 (2015).
12. Clancy KW, Weerapana E, Thompson PR. Detection and identification of protein citrullination in complex biological systems. *Curr Opin Chem Biol* **30**, 1-6 (2016).
13. Verheul MK, *et al.* Pitfalls in the detection of citrullination and carbamylation. *Autoimmun Rev* **17**, 136-141 (2018).
14. Yang ML, Sodre FMC, Mamula MJ, Overbergh L. Citrullination and PAD Enzyme Biology in Type 1 Diabetes - Regulators of Inflammation, Autoimmunity, and Pathology. *Front Immunol* **12**, 678953 (2021).
15. Baekkeskov S, *et al.* Identification of the 64K autoantigen in insulin-dependent diabetes as the GABA-synthesizing enzyme glutamic acid decarboxylase. *Nature* **347**, 151-156 (1990).
16. Pihoker C, Gilliam LK, Hampe CS, Lernmark A. Autoantibodies in diabetes. *Diabetes* **54 Suppl 2**, S52-61 (2005).
17. Dade M, *et al.* Neurological Syndromes Associated with Anti-GAD Antibodies. *Int J Mol Sci* **21**, (2020).
18. Arbuckle MR, *et al.* Development of autoantibodies before the clinical onset of systemic lupus erythematosus. *N Engl J Med* **349**, 1526-1533 (2003).
19. Guo YP, *et al.* The prevalence of antinuclear antibodies in the general population of china: a cross-sectional study. *Curr Ther Res Clin Exp* **76**, 116-119 (2014).

20. Akmatov MK, *et al.* Anti-nuclear autoantibodies in the general German population: prevalence and lack of association with selected cardiovascular and metabolic disorders-findings of a multicenter population-based study. *Arthritis Res Ther* **19**, 127 (2017).
21. Pisetsky DS. Antinuclear antibody testing - misunderstood or misbegotten? *Nat Rev Rheumatol* **13**, 495-502 (2017).
22. Lac P, Saunders S, Tutunea-Fatan E, Barra L, Bell DA, Cairns E. Immune responses to peptides containing homocitrulline or citrulline in the DR4-transgenic mouse model of rheumatoid arthritis. *J Autoimmun* **89**, 75-81 (2018).
23. Ireland J, Herzog J, Unanue ER. Cutting edge: unique T cells that recognize citrullinated peptides are a feature of protein immunization. *J Immunol* **177**, 1421-1425 (2006).

REVIEWER COMMENTS

Reviewer #1 (Remarks to the Author):

The authors have addressed all my concerns/suggestions.

Reviewer #2 (Remarks to the Author):

The authors have adequately addressed my concerns.